# MixMatch: A Holistic Approach to Semi-Supervised Learning

**David Berthelot**
Google Research
dberth@google.com

**Nicholas Carlini**
Google Research
ncarlini@google.com

**Ian Goodfellow**
Work done at Google
ian-academic@mailfence.com

**Avital Oliver**
Google Research
avitalo@google.com

**Nicolas Papernot**
Google Research
papernot@google.com

**Colin Raffel**
Google Research
craffel@google.com

## Abstract

Semi-supervised learning has proven to be a powerful paradigm for leveraging unlabeled data to mitigate the reliance on large labeled datasets. In this work, we unify the current dominant approaches for semi-supervised learning to produce a new algorithm, MixMatch, that guesses low-entropy labels for data-augmented unlabeled examples and mixes labeled and unlabeled data using MixUp. MixMatch obtains state-of-the-art results by a large margin across many datasets and labeled data amounts. For example, on CIFAR-10 with 250 labels, we reduce error rate by a factor of 4 (from $38\%$ to $11\%$) and by a factor of 2 on STL-10. We also demonstrate how MixMatch can help achieve a dramatically better accuracy-privacy trade-off for differential privacy. Finally, we perform an ablation study to tease apart which components of MixMatch are most important for its success. We release all code used in our experiments.[1]

## 1 Introduction

Much of the recent success in training large, deep neural networks is thanks in part to the existence of large labeled datasets. Yet, collecting labeled data is expensive for many learning tasks because it necessarily involves expert knowledge. This is perhaps best illustrated by medical tasks where measurements call for expensive machinery and labels are the fruit of a time-consuming analysis that draws from multiple human experts. Furthermore, data labels may contain private information. In comparison, in many tasks it is much easier or cheaper to obtain unlabeled data.

Semi-supervised learning [6] (SSL) seeks to largely alleviate the need for labeled data by allowing a model to leverage unlabeled data. Many recent approaches for semi-supervised learning add a loss term which is computed on unlabeled data and encourages the model to generalize better to unseen data. In much recent work, this loss term falls into one of three classes (discussed further in Section 2): entropy minimization [18, 28]—which encourages the model to output confident predictions on unlabeled data; consistency regularization—which encourages the model to produce the same output distribution when its inputs are perturbed; and generic regularization—which encourages the model to generalize well and avoid overfitting the training data.

In this paper, we introduce MixMatch, an SSL algorithm which introduces a single loss that gracefully unifies these dominant approaches to semi-supervised learning. Unlike previous methods, MixMatch targets all the properties at once which we find leads to the following benefits:

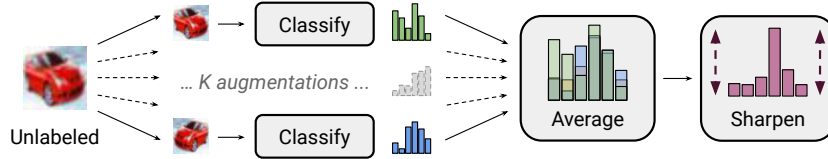

Figure 1: Diagram of the label guessing process used in $\mathrm{MixMatch}$. Stochastic data augmentation is applied to an unlabeled image $K$ times, and each augmented image is fed through the classifier. Then, the average of these $K$ predictions is "sharpened" by adjusting the distribution's temperature. See algorithm 1 for a full description.

- Experimentally, we show that $\mathrm{MixMatch}$ obtains state-of-the-art results on all standard image benchmarks (section 4.2), and reducing the error rate on CIFAR-10 by a factor of 4;
- We further show in an ablation study that $\mathrm{MixMatch}$ is greater than the sum of its parts;
- We demonstrate in section 4.3 that $\mathrm{MixMatch}$ is useful for differentially private learning, enabling students in the PATE framework [36] to obtain new state-of-the-art results that simultaneously strengthen both privacy guarantees and accuracy.

In short, $\mathrm{MixMatch}$ introduces a unified loss term for unlabeled data that seamlessly reduces entropy while maintaining consistency and remaining compatible with traditional regularization techniques.

## 2   Related Work

To set the stage for $\mathrm{MixMatch}$, we first introduce existing methods for SSL. We focus mainly on those which are currently state-of-the-art and that $\mathrm{MixMatch}$ builds on; there is a wide literature on SSL techniques that we do not discuss here (e.g., "transductive" models [14, 22, 21], graph-based methods [49, 4, 29], generative modeling [3, 27, 41, 9, 17, 23, 38, 34, 42], etc.). More comprehensive overviews are provided in [49, 6]. In the following, we will refer to a generic model $\mathrm{p_{model}}(y \mid x; \theta)$ which produces a distribution over class labels $y$ for an input $x$ with parameters $\theta$.

### 2.1   Consistency Regularization

A common regularization technique in supervised learning is *data augmentation*, which applies input transformations assumed to leave class semantics unaffected. For example, in image classification, it is common to elastically deform or add noise to an input image, which can dramatically change the pixel content of an image without altering its label [7, 43, 10]. Roughly speaking, this can artificially expand the size of a training set by generating a near-infinite stream of new, modified data. *Consistency regularization* applies data augmentation to semi-supervised learning by leveraging the idea that a classifier should output the same class distribution for an unlabeled example even after it has been augmented. More formally, consistency regularization enforces that an unlabeled example $x$ should be classified the same as $\mathrm{Augment}(x)$, an augmentation of itself.

In the simplest case, for unlabeled points $x$, prior work [25, 40] adds the loss term

$$\|\mathrm{p_{model}}(y \mid \mathrm{Augment}(x); \theta) - \mathrm{p_{model}}(y \mid \mathrm{Augment}(x); \theta)\|_2^2. \tag{1}$$

Note that $\mathrm{Augment}(x)$ is a stochastic transformation, so the two terms in eq. (1) are not identical. "Mean Teacher" [44] replaces one of the terms in eq. (1) with the output of the model using an exponential moving average of model parameter values. This provides a more stable target and was found empirically to significantly improve results. A drawback to these approaches is that they use domain-specific data augmentation strategies. "Virtual Adversarial Training" [31] (VAT) addresses this by instead computing an additive perturbation to apply to the input which maximally changes the output class distribution. MixMatch utilizes a form of consistency regularization through the use of standard data augmentation for images (random horizontal flips and crops).

### 2.2   Entropy Minimization

A common underlying assumption in many semi-supervised learning methods is that the classifier's decision boundary should not pass through high-density regions of the marginal data distribution.

One way to enforce this is to require that the classifier output low-entropy predictions on unlabeled data. This is done explicitly in [18] with a loss term which minimizes the entropy of $\mathrm{p}_{\mathrm{model}}(y \mid x; \theta)$ for unlabeled data $x$. This form of entropy minimization was combined with VAT in [31] to obtain stronger results. "Pseudo-Label" [28] does entropy minimization implicitly by constructing hard (1-hot) labels from high-confidence predictions on unlabeled data and using these as training targets in a standard cross-entropy loss. MixMatch also implicitly achieves entropy minimization through the use of a "sharpening" function on the target distribution for unlabeled data, described in section 3.2.

## 2.3 Traditional Regularization

Regularization refers to the general approach of imposing a constraint on a model to make it harder to memorize the training data and therefore hopefully make it generalize better to unseen data [19]. We use weight decay which penalizes the $L_2$ norm of the model parameters [30, 46]. We also use $\mathrm{MixUp}$ [47] in $\mathrm{MixMatch}$ to encourage convex behavior "between" examples. We utilize $\mathrm{MixUp}$ as both as a regularizer (applied to labeled datapoints) and a semi-supervised learning method (applied to unlabeled datapoints). $\mathrm{MixUp}$ has been previously applied to semi-supervised learning; in particular, the concurrent work of [45] uses a subset of the methodology used in MixMatch. We clarify the differences in our ablation study (section 4.2.3).

## 3 MixMatch

In this section, we introduce $\mathrm{MixMatch}$, our proposed semi-supervised learning method. $\mathrm{MixMatch}$ is a "holistic" approach which incorporates ideas and components from the dominant paradigms for SSL discussed in section 2. Given a batch $\mathcal{X}$ of labeled examples with one-hot targets (representing one of $L$ possible labels) and an equally-sized batch $\mathcal{U}$ of unlabeled examples, $\mathrm{MixMatch}$ produces a processed batch of augmented labeled examples $\mathcal{X}'$ and a batch of augmented unlabeled examples with "guessed" labels $\mathcal{U}'$. $\mathcal{U}'$ and $\mathcal{X}'$ are then used in computing separate labeled and unlabeled loss terms. More formally, the combined loss $\mathcal{L}$ for semi-supervised learning is defined as

$$\mathcal{X}',\mathcal{U}' = \mathrm{MixMatch}(\mathcal{X},\mathcal{U},T,K,\alpha) \tag{2}$$

$$\mathcal{L}_{\mathcal{X}} = \frac{1}{|\mathcal{X}'|} \sum_{x,p\in\mathcal{X}'} \mathrm{H}(p, \mathrm{p}_{\mathrm{model}}(y \mid x; \theta)) \tag{3}$$

$$\mathcal{L}_{\mathcal{U}} = \frac{1}{L|\mathcal{U}'|} \sum_{u,q\in\mathcal{U}'} \|q - \mathrm{p}_{\mathrm{model}}(y \mid u; \theta)\|_2^2 \tag{4}$$

$$\mathcal{L} = \mathcal{L}_{\mathcal{X}} + \lambda_{\mathcal{U}}\mathcal{L}_{\mathcal{U}} \tag{5}$$

where $\mathrm{H}(p, q)$ is the cross-entropy between distributions $p$ and $q$, and $T$, $K$, $\alpha$, and $\lambda_{\mathcal{U}}$ are hyperparameters described below. The full $\mathrm{MixMatch}$ algorithm is provided in algorithm 1, and a diagram of the label guessing process is shown in fig. 1. Next, we describe each part of $\mathrm{MixMatch}$.

## 3.1 Data Augmentation

As is typical in many SSL methods, we use data augmentation both on labeled and unlabeled data. For each $x_b$ in the batch of labeled data $\mathcal{X}$, we generate a transformed version $\hat{x}_b = \mathrm{Augment}(x_b)$ (algorithm 1, line 3). For each $u_b$ in the batch of unlabeled data $\mathcal{U}$, we generate $K$ augmentations $\hat{u}_{b,k} = \mathrm{Augment}(u_b), k \in (1,\dots,K)$ (algorithm 1, line 5). We use these individual augmentations to generate a "guessed label" $q_b$ for each $u_b$, through a process we describe in the following subsection.

## 3.2 Label Guessing

For each unlabeled example in $\mathcal{U}$, $\mathrm{MixMatch}$ produces a "guess" for the example's label using the model's predictions. This guess is later used in the unsupervised loss term. To do so, we compute the average of the model's predicted class distributions across all the $K$ augmentations of $u_b$ by

$$\bar{q}_b = \frac{1}{K} \sum_{k=1}^{K} \mathrm{p}_{\mathrm{model}}(y \mid \hat{u}_{b,k}; \theta) \tag{6}$$

in algorithm 1, line 7. Using data augmentation to obtain an artificial target for an unlabeled example is common in consistency regularization methods [25, 40, 44].

**Algorithm 1** MixMatch takes a batch of labeled data $\mathcal{X}$ and a batch of unlabeled data $\mathcal{U}$ and produces a collection $\mathcal{X}'$ (resp. $\mathcal{U}'$) of processed labeled examples (resp. unlabeled with guessed labels).

---

1: **Input:** Batch of labeled examples and their one-hot labels $\mathcal{X} = \big((x_b, p_b); b \in (1, \ldots, B)\big)$, batch of unlabeled examples $\mathcal{U} = \big(u_b; b \in (1, \ldots, B)\big)$, sharpening temperature $T$, number of augmentations $K$, Beta distribution parameter $\alpha$ for MixUp.
2: **for** $b = 1$ **to** $B$ **do**
3:     $\hat{x}_b = \text{Augment}(x_b)$    // *Apply data augmentation to $x_b$*
4:     **for** $k = 1$ **to** $K$ **do**
5:         $\hat{u}_{b,k} = \text{Augment}(u_b)$    // *Apply $k^{th}$ round of data augmentation to $u_b$*
6:     **end for**
7:     $\bar{q}_b = \frac{1}{K} \sum_k \text{p}_{\text{model}}(y \mid \hat{u}_{b,k}; \theta)$    // *Compute average predictions across all augmentations of $u_b$*
8:     $q_b = \text{Sharpen}(\bar{q}_b, T)$    // *Apply temperature sharpening to the average prediction (see eq. (7))*
9: **end for**
10: $\hat{\mathcal{X}} = \big((\hat{x}_b, p_b); b \in (1, \ldots, B)\big)$    // *Augmented labeled examples and their labels*
11: $\hat{\mathcal{U}} = \big((\hat{u}_{b,k}, q_b); b \in (1, \ldots, B), k \in (1, \ldots, K)\big)$    // *Augmented unlabeled examples, guessed labels*
12: $\mathcal{W} = \text{Shuffle}\big(\text{Concat}(\hat{\mathcal{X}}, \hat{\mathcal{U}})\big)$    // *Combine and shuffle labeled and unlabeled data*
13: $\mathcal{X}' = \big(\text{MixUp}(\hat{\mathcal{X}}_i, \mathcal{W}_i); i \in (1, \ldots, |\hat{\mathcal{X}}|)\big)$    // *Apply* MixUp *to labeled data and entries from $\mathcal{W}$*
14: $\mathcal{U}' = \big(\text{MixUp}(\hat{\mathcal{U}}_i, \mathcal{W}_{i+|\hat{\mathcal{X}}|}); i \in (1, \ldots, |\hat{\mathcal{U}}|)\big)$    // *Apply* MixUp *to unlabeled data and the rest of $\mathcal{W}$*
15: **return** $\mathcal{X}', \mathcal{U}'$

---

**Sharpening.** In generating a label guess, we perform one additional step inspired by the success of entropy minimization in semi-supervised learning (discussed in section 2.2). Given the average prediction over augmentations $\bar{q}_b$, we apply a sharpening function to reduce the entropy of the label distribution. In practice, for the sharpening function, we use the common approach of adjusting the "temperature" of this categorical distribution [16], which is defined as the operation

$$\text{Sharpen}(p, T)_i := p_i^{\frac{1}{T}} \Big/ \sum_{j=1}^{L} p_j^{\frac{1}{T}} \tag{7}$$

where $p$ is some input categorical distribution (specifically in MixMatch, $p$ is the average class prediction over augmentations $\bar{q}_b$, as shown in algorithm 1, line 8) and $T$ is a hyperparameter. As $T \to 0$, the output of $\text{Sharpen}(p, T)$ will approach a Dirac ("one-hot") distribution. Since we will later use $q_b = \text{Sharpen}(\bar{q}_b, T)$ as a target for the model's prediction for an augmentation of $u_b$, lowering the temperature encourages the model to produce lower-entropy predictions.

### 3.3 MixUp

We use MixUp for semi-supervised learning, and unlike past work for SSL we mix both labeled examples and unlabeled examples with label guesses (generated as described in section 3.2). To be compatible with our separate loss terms, we define a slightly modified version of MixUp. For a pair of two examples with their corresponding labels probabilities $(x_1, p_1), (x_2, p_2)$ we compute $(x', p')$ by

$$\lambda \sim \text{Beta}(\alpha, \alpha) \tag{8}$$

$$\lambda' = \max(\lambda, 1 - \lambda) \tag{9}$$

$$x' = \lambda' x_1 + (1 - \lambda') x_2 \tag{10}$$

$$p' = \lambda' p_1 + (1 - \lambda') p_2 \tag{11}$$

where $\alpha$ is a hyperparameter. Vanilla MixUp omits eq. (9) (i.e. it sets $\lambda' = \lambda$). Given that labeled and unlabeled examples are concatenated in the same batch, we need to preserve the order of the batch to compute individual loss components appropriately. This is achieved by eq. (9) which ensures that $x'$ is closer to $x_1$ than to $x_2$. To apply MixUp, we first collect all augmented labeled examples with their labels and all unlabeled examples with their guessed labels into

$$\hat{\mathcal{X}} = \big((\hat{x}_b, p_b); b \in (1, \ldots, B)\big) \tag{12}$$

$$\hat{\mathcal{U}} = \big((\hat{u}_{b,k}, q_b); b \in (1, \ldots, B), k \in (1, \ldots, K)\big) \tag{13}$$

(algorithm 1, lines 10–11). Then, we combine these collections and shuffle the result to form $\mathcal{W}$ which will serve as a data source for $\mathrm{MixUp}$ (algorithm 1, line 12). For each the $i^{th}$ example-label pair in $\hat{\mathcal{X}}$, we compute $\mathrm{MixUp}(\hat{\mathcal{X}}_i, \mathcal{W}_i)$ and add the result to the collection $\mathcal{X}'$ (algorithm 1, line 13). We compute $\mathcal{U}'_i = \mathrm{MixUp}(\hat{\mathcal{U}}_i, \mathcal{W}_{i+|\hat{\mathcal{X}}|})$ for $i \in (1, \ldots, |\hat{\mathcal{U}}|)$, intentionally using the remainder of $\mathcal{W}$ that was not used in the construction of $\mathcal{X}'$ (algorithm 1, line 14). To summarize, $\mathrm{MixMatch}$ transforms $\mathcal{X}$ into $\mathcal{X}'$, a collection of labeled examples which have had data augmentation and $\mathrm{MixUp}$ (potentially mixed with an unlabeled example) applied. Similarly, $\mathcal{U}$ is transformed into $\mathcal{U}'$, a collection of multiple augmentations of each unlabeled example with corresponding label guesses.

### 3.4 Loss Function

Given our processed batches $\mathcal{X}'$ and $\mathcal{U}'$, we use the standard semi-supervised loss shown in eqs. (3) to (5). Equation (5) combines the typical cross-entropy loss between labels and model predictions from $\mathcal{X}'$ with the squared $L_2$ loss on predictions and guessed labels from $\mathcal{U}'$. We use this $L_2$ loss in eq. (4) (the multiclass Brier score [5]) because, unlike the cross-entropy, it is bounded and less sensitive to incorrect predictions. For this reason, it is often used as the unlabeled data loss in SSL [25, 44] as well as a measure of predictive uncertainty [26]. We do not propagate gradients through computing the guessed labels, as is standard [25, 44, 31, 35]

### 3.5 Hyperparameters

Since $\mathrm{MixMatch}$ combines multiple mechanisms for leveraging unlabeled data, it introduces various hyperparameters – specifically, the sharpening temperature $T$, number of unlabeled augmentations $K$, $\alpha$ parameter for $\mathrm{Beta}$ in $\mathrm{MixUp}$, and the unsupervised loss weight $\lambda_{\mathcal{U}}$. In practice, semi-supervised learning methods with many hyperparameters can be problematic because cross-validation is difficult with small validation sets [35, 39, 35]. However, we find in practice that most of $\mathrm{MixMatch}$'s hyperparameters can be fixed and do not need to be tuned on a per-experiment or per-dataset basis. Specifically, for all experiments we set $T = 0.5$ and $K = 2$. Further, we only change $\alpha$ and $\lambda_{\mathcal{U}}$ on a per-dataset basis; we found that $\alpha = 0.75$ and $\lambda_{\mathcal{U}} = 100$ are good starting points for tuning. In all experiments, we linearly ramp up $\lambda_{\mathcal{U}}$ to its maximum value over the first 16,000 steps of training as is common practice [44].

## 4 Experiments

We test the effectiveness of $\mathrm{MixMatch}$ on standard SSL benchmarks (section 4.2). Our ablation study teases apart the contribution of each of $\mathrm{MixMatch}$'s components (section 4.2.3). As an additional application, we consider privacy-preserving learning in section 4.3.

### 4.1 Implementation details

Unless otherwise noted, in all experiments we use the "Wide ResNet-28" model from [35]. Our implementation of the model and training procedure closely matches that of [35] (including using 5000 examples to select the hyperparameters), except for the following differences: First, instead of decaying the learning rate, we evaluate models using an exponential moving average of their parameters with a decay rate of 0.999. Second, we apply a weight decay of 0.0004 at each update for the Wide ResNet-28 model. Finally, we checkpoint every $2^{16}$ training samples and report the median error rate of the last 20 checkpoints. This simplifies the analysis at a potential cost to accuracy by, for example, averaging checkpoints [2] or choosing the checkpoint with the lowest validation error.

### 4.2 Semi-Supervised Learning

First, we evaluate the effectiveness of $\mathrm{MixMatch}$ on four standard benchmark datasets: CIFAR-10 and CIFAR-100 [24], SVHN [32], and STL-10 [8]. Standard practice for evaluating semi-supervised learning on the first three datasets is to treat most of the dataset as unlabeled and use a small portion as labeled data. STL-10 is a dataset specifically designed for SSL, with 5,000 labeled images and 100,000 unlabeled images which are drawn from a slightly different distribution than the labeled data.

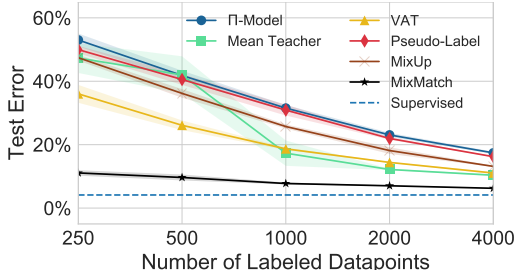
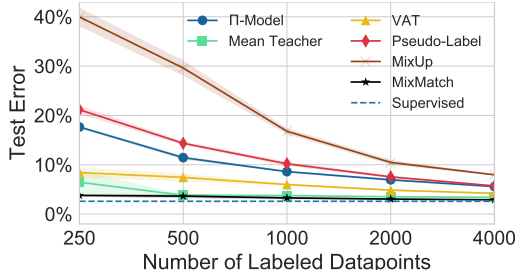

Figure 2: Error rate comparison of MixMatch to baseline methods on CIFAR-10 for a varying number of labels. Exact numbers are provided in table 5 (appendix). "Supervised" refers to training with all 50000 training examples and no unlabeled data. With 250 labels MixMatch reaches an error rate comparable to next-best method's performance with 4000 labels.

Figure 3: Error rate comparison of MixMatch to baseline methods on SVHN for a varying number of labels. Exact numbers are provided in table 6 (appendix). "Supervised" refers to training with all 73257 training examples and no unlabeled data. With 250 examples MixMatch nearly reaches the accuracy of supervised training for this model.

### 4.2.1 Baseline Methods

As baselines, we consider the four methods considered in [35] (Π-Model [25, 40], Mean Teacher [44], Virtual Adversarial Training [31], and Pseudo-Label [28]) which are described in section 2. We also use MixUp [47] on its own as a baseline. MixUp is designed as a regularizer for supervised learning, so we modify it for SSL by applying it both to augmented labeled examples and augmented unlabeled examples with their corresponding predictions. In accordance with standard usage of MixUp, we use a cross-entropy loss between the MixUp-generated guess label and the model's prediction. As advocated by [35], we reimplemented each of these methods in the same codebase and applied them to the same model (described in section 4.1) to ensure a fair comparison. We re-tuned the hyperparameters for each baseline method, which generally resulted in a marginal accuracy improvement compared to those in [35], thereby providing a more competitive experimental setting for testing out MixMatch.

### 4.2.2 Results

**CIFAR-10** For CIFAR-10, we evaluate the accuracy of each method with a varying number of labeled examples from 250 to 4000 (as is standard practice). The results can be seen in fig. 2. We used $\lambda_{\mathcal{U}} = 75$ for CIFAR-10. We created 5 splits for each number of labeled points, each with a different random seed. Each model was trained on each split and the error rates were reported by the mean and variance across splits. We find that MixMatch outperforms all other methods by a significant margin, for example reaching an error rate of 6.24% with 4000 labels. For reference, on the same model, fully supervised training on all 50000 samples achieves an error rate of 4.17%. Furthermore, MixMatch obtains an error rate of 11.08% with only 250 labels. For comparison, at 250 labels the next-best-performing method (VAT [31]) achieves an error rate of 36.03, over $4.5\times$ higher than MixMatch considering that 4.17% is the error limit obtained on our model with fully supervised learning. In addition, at 4000 labels the next-best-performing method (Mean Teacher [44]) obtains an error rate of 10.36%, which suggests that MixMatch can achieve similar performance with only 1/16 as many labels. We believe that the most interesting comparisons are with very few labeled data points since it reveals the method's sample efficiency which is central to SSL.

**CIFAR-10 and CIFAR-100 with a larger model** Some prior work [44, 2] has also considered the use of a larger, 26 million-parameter model. Our base model, as used in [35], has only 1.5 million parameters which confounds comparison with these results. For a more reasonable comparison to these results, we measure the effect of increasing the width of our base ResNet model and evaluate MixMatch's performance on a 28-layer Wide Resnet model which has 135 filters per layer, resulting in 26 million parameters. We also evaluate MixMatch on this larger model on CIFAR-100 with 10000 labels, to compare to the corresponding result from [2]. The results are shown in table 1. In general, MixMatch matches or outperforms the best results from [2], though we note that the comparison still remains problematic due to the fact that the model from [44, 2] also uses more

| Method | CIFAR-10 | CIFAR-100 |
|---|---|---|
| Mean Teacher [44] | 6.28 | - |
| SWA [2] | 5.00 | 28.80 |
| MixMatch | $4.95 \pm 0.08$ | $25.88 \pm 0.30$ |

Table 1: CIFAR-10 and CIFAR-100 error rate (with $4{,}000$ and $10{,}000$ labels respectively) with larger models (26 million parameters).

| Method | 1000 labels | 5000 labels |
|---|---|---|
| CutOut [12] | - | 12.74 |
| IIC [20] | - | 11.20 |
| SWWAE [48] | 25.70 | - |
| CC-GAN$^2$ [11] | 22.20 | - |
| MixMatch | $10.18 \pm 1.46$ | 5.59 |

Table 2: STL-10 error rate using 1000-label splits or the entire 5000-label training set.

| Labels | 250 | 500 | 1000 | 2000 | 4000 | All |
|---|---|---|---|---|---|---|
| SVHN | $3.78 \pm 0.26$ | $3.64 \pm 0.46$ | $3.27 \pm 0.31$ | $3.04 \pm 0.13$ | $2.89 \pm 0.06$ | 2.59 |
| SVHN+Extra | $2.22 \pm 0.08$ | $2.17 \pm 0.07$ | $2.18 \pm 0.06$ | $2.12 \pm 0.03$ | $2.07 \pm 0.05$ | 1.71 |

Table 3: Comparison of error rates for SVHN and SVHN+Extra for $\mathrm{MixMatch}$. The last column ("All") contains the fully-supervised performance with all labels in the corresponding training set.

sophisticated "shake-shake" regularization [15]. For this model, we used a weight decay of $0.0008$. We used $\lambda_{\mathcal{U}} = 75$ for CIFAR-10 and $\lambda_{\mathcal{U}} = 150$ for CIFAR-100.

**SVHN and SVHN+Extra** As with CIFAR-10, we evaluate the performance of each SSL method on SVHN with a varying number of labels from 250 to 4000. As is standard practice, we first consider the setting where the 73257-example training set is split into labeled and unlabeled data. The results are shown in fig. 3. We used $\lambda_{\mathcal{U}} = 250$. Here again the models were evaluated on 5 splits for each number of labeled points, each with a different random seed. We found $\mathrm{MixMatch}$'s performance to be relatively constant (and better than all other methods) across all amounts of labeled data. Surprisingly, after additional tuning we were able to obtain extremely good performance from Mean Teacher [44], though its error rate was consistently slightly higher than $\mathrm{MixMatch}$'s.

Note that SVHN has two training sets: *train* and *extra*. In fully-supervised learning, both sets are concatenated to form the full training set (604388 samples). In SSL, for historical reasons the *extra* set was left aside and only train was used (73257 samples). We argue that leveraging both *train* and *extra* for the unlabeled data is more interesting since it exhibits a higher ratio of unlabeled samples over labeled ones. We report error rates for both SVHN and SVHN+Extra in table 3. For SVHN+Extra we used $\alpha = 0.25, \lambda_{\mathcal{U}} = 250$ and a lower weight decay of $0.000002$ due to the larger amount of available data. We found that on both training sets, $\mathrm{MixMatch}$ nearly matches the fully-supervised performance on the same training set almost immediately – for example, $\mathrm{MixMatch}$ achieves an error rate of $2.22\%$ with only 250 labels on SVHN+Extra compared to the fully-supervised performance of $1.71\%$. Interestingly, on SVHN+Extra $\mathrm{MixMatch}$ outperformed fully supervised training on SVHN without *extra* ($2.59\%$ error) for every labeled data amount considered. To emphasize the importance of this, consider the following scenario: You have 73257 examples from SVHN with 250 examples labeled and are given a choice: You can either obtain $8\times$ more unlabeled data and use $\mathrm{MixMatch}$ or obtain $293\times$ more labeled data and use fully-supervised learning. Our results suggest that obtaining additional unlabeled data and using $\mathrm{MixMatch}$ is more effective, which conveniently is likely much cheaper than obtaining $293\times$ more labels.

**STL-10** STL-10 contains 5000 training examples aimed at being used with 10 predefined folds (we use the first 5 only) with 1000 examples each. However, some prior work trains on all 5000 examples. We thus compare in both experimental settings. With 1000 examples $\mathrm{MixMatch}$ surpasses both the state-of-the-art for 1000 examples as well as the state-of-the-art using all 5000 labeled examples. Note that none of the baselines in table 2 use the same experimental setup (i.e. model), so it is difficult to directly compare the results; however, because $\mathrm{MixMatch}$ obtains the lowest error by a factor of two, we take this to be a vote in confidence of our method. We used $\lambda_{\mathcal{U}} = 50$.

### 4.2.3 Ablation Study

Since $\mathrm{MixMatch}$ combines various semi-supervised learning mechanisms, it has a good deal in common with existing methods in the literature. As a result, we study the effect of removing or

| Ablation | 250 labels | 4000 labels |
|---|---|---|
| MixMatch | 11.80 | 6.00 |
| MixMatch without distribution averaging ($K = 1$) | 17.09 | 8.06 |
| MixMatch with $K = 3$ | 11.55 | 6.23 |
| MixMatch with $K = 4$ | 12.45 | 5.88 |
| MixMatch without temperature sharpening ($T = 1$) | 27.83 | 10.59 |
| MixMatch with parameter EMA | 11.86 | 6.47 |
| MixMatch without MixUp | 39.11 | 10.97 |
| MixMatch with MixUp on labeled only | 32.16 | 9.22 |
| MixMatch with MixUp on unlabeled only | 12.35 | 6.83 |
| MixMatch with MixUp on separate labeled and unlabeled | 12.26 | 6.50 |
| Interpolation Consistency Training [45] | 38.60 | 6.81 |

Table 4: Ablation study results. All values are error rates on CIFAR-10 with 250 or 4000 labels.

adding components in order to provide additional insight into what makes MixMatch performant. Specifically, we measure the effect of

- using the mean class distribution over $K$ augmentations or using the class distribution for a single augmentation (i.e. setting $K = 1$)

- removing temperature sharpening (i.e. setting $T = 1$)

- using an exponential moving average (EMA) of model parameters when producing guessed labels, as is done by Mean Teacher [44]

- performing MixUp between labeled examples only, unlabeled examples only, and without mixing across labeled and unlabeled examples

- using Interpolation Consistency Training [45], which can be seen as a special case of this ablation study where only unlabeled mixup is used, no sharpening is applied and EMA parameters are used for label guessing.

We carried out the ablation on CIFAR-10 with 250 and 4000 labels; the results are shown in table 4. We find that each component contributes to MixMatch's performance, with the most dramatic differences in the 250-label setting. Despite Mean Teacher's effectiveness on SVHN (fig. 3), we found that using a similar EMA of parameter values hurt MixMatch's performance slightly.

## 4.3 Privacy-Preserving Learning and Generalization

Learning with privacy allows us to measure our approach's ability to generalize. Indeed, protecting the privacy of training data amounts to proving that the model does not overfit: a learning algorithm is said to be differentially private (the most widely accepted technical definition of privacy) if adding, modifying, or removing any of its training samples is guaranteed not to result in a statistically significant difference in the model parameters learned [13]. For this reason, learning with differential privacy is, in practice, a form of regularization [33]. Each training data access constitutes a potential privacy leakage, encoded as the pair of the input and its label. Hence, approaches for deep learning from private training data, such as DP-SGD [1] and PATE [36], benefit from accessing as few labeled private training points as possible when computing updates to the model parameters. Semi-supervised learning is a natural fit for this setting.

We use the PATE framework for learning with privacy. A student is trained in a semi-supervised way from public *unlabeled* data, part of which is labeled by an ensemble of teachers with access to private *labeled* training data. The fewer labels a student requires to reach a fixed accuracy, the stronger is the privacy guarantee it provides. Teachers use a noisy voting mechanism to respond to label queries from the student, and they may choose *not* to provide a label when they cannot reach a sufficiently strong consensus. For this reason, if MixMatch improves the performance of PATE, it would also illustrate MixMatch's improved generalization from few canonical exemplars of each class.

We compare the accuracy-privacy trade-off achieved by MixMatch to a VAT [31] baseline on SVHN. VAT achieved the previous state-of-the-art of $91.6\%$ test accuracy for a privacy loss of $\varepsilon = 4.96$ [37]. Because MixMatch performs well with few labeled points, it is able to achieve $95.21 \pm 0.17\%$ test

accuracy for a much smaller privacy loss of $\varepsilon = 0.97$. Because $e^\varepsilon$ is used to measure the degree of privacy, the improvement is approximately $e^4 \approx 55\times$, a significant improvement. A privacy loss $\varepsilon$ below 1 corresponds to a much stronger privacy guarantee. Note that in the private training setting the student model only uses 10,000 total examples.

## 5 Conclusion

We introduced $\mathrm{MixMatch}$, a semi-supervised learning method which combines ideas and components from the current dominant paradigms for SSL. Through extensive experiments on semi-supervised and privacy-preserving learning, we found that $\mathrm{MixMatch}$ exhibited significantly improved performance compared to other methods in all settings we studied, often by a factor of two or more reduction in error rate. In future work, we are interested in incorporating additional ideas from the semi-supervised learning literature into hybrid methods and continuing to explore which components result in effective algorithms. Separately, most modern work on semi-supervised learning algorithms is evaluated on image benchmarks; we are interested in exploring the effectiveness of $\mathrm{MixMatch}$ in other domains.

**Acknowledgement**

We would like to thank Balaji Lakshminarayanan for his helpful theoretical insights.

## Footnotes

[1] https://github.com/google-research/mixmatch

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
