[Supplementary Material]



# A   Notation and definitions

| Notation | Definition |
|---|---|
| $\mathrm{H}(p, q)$ | Cross-entropy between "target" distribution $p$ and "predicted" distribution $q$ |
| $x$ | A labeled example, used as input to a model |
| $p$ | A (one-hot) label |
| $L$ | The number of possible label classes (the dimensionality of $p$) |
| $\mathcal{X}$ | A batch of labeled examples and their labels |
| $\mathcal{X}'$ | A batch of processed labeled examples produced by MixMatch |
| $u$ | An unlabeled example, used as input to a model |
| $q$ | A guessed label distribution for an unlabeled example |
| $\mathcal{U}$ | A batch of unlabeled examples |
| $\mathcal{U}'$ | A batch of processed unlabeled examples with their label guesses produced by MixMatch |
| $\theta$ | The model's parameters |
| $\mathrm{p_{model}}(y \mid x; \theta)$ | The model's predicted distribution over classes |
| $\mathrm{Augment}(x)$ | A stochastic data augmentation function that returns a modified version of $x$. For example, $\mathrm{Augment}(\cdot)$ could implement randomly shifting an input image, or implement adding a perturbation sampled from a Gaussian distribution to $x$. |
| $\lambda_{\mathcal{U}}$ | A hyper-parameter weighting the contribution of the unlabeled examples to the training loss |
| $\alpha$ | Hyperparameter for the Beta distribution used in MixUp |
| $T$ | Temperature parameter for sharpening used in MixMatch |
| $K$ | Number of augmentations used when guessing labels in MixMatch |

# B Tabular results

## B.1 CIFAR-10

Training the same model with supervised learning on the entire 50000-example training set achieved an error rate of 4.13%.

| Methods/Labels | 250 | 500 | 1000 | 2000 | 4000 |
|---|---|---|---|---|---|
| PiModel | $53.02 \pm 2.05$ | $41.82 \pm 1.52$ | $31.53 \pm 0.98$ | $23.07 \pm 0.66$ | $17.41 \pm 0.37$ |
| PseudoLabel | $49.98 \pm 1.17$ | $40.55 \pm 1.70$ | $30.91 \pm 1.73$ | $21.96 \pm 0.42$ | $16.21 \pm 0.11$ |
| Mixup | $47.43 \pm 0.92$ | $36.17 \pm 1.36$ | $25.72 \pm 0.66$ | $18.14 \pm 1.06$ | $13.15 \pm 0.20$ |
| VAT | $36.03 \pm 2.82$ | $26.11 \pm 1.52$ | $18.68 \pm 0.40$ | $14.40 \pm 0.15$ | $11.05 \pm 0.31$ |
| MeanTeacher | $47.32 \pm 4.71$ | $42.01 \pm 5.86$ | $17.32 \pm 4.00$ | $12.17 \pm 0.22$ | $10.36 \pm 0.25$ |
| MixMatch | $11.08 \pm 0.87$ | $9.65 \pm 0.94$ | $7.75 \pm 0.32$ | $7.03 \pm 0.15$ | $6.24 \pm 0.06$ |

Table 5: Error rate (%) for CIFAR10.

## B.2 SVHN

Training the same model with supervised learning on the entire 73257-example training set achieved an error rate of 2.59%.

| Methods/Labels | 250 | 500 | 1000 | 2000 | 4000 |
|---|---|---|---|---|---|
| PiModel | $17.65 \pm 0.27$ | $11.44 \pm 0.39$ | $8.60 \pm 0.18$ | $6.94 \pm 0.27$ | $5.57 \pm 0.14$ |
| PseudoLabel | $21.16 \pm 0.88$ | $14.35 \pm 0.37$ | $10.19 \pm 0.41$ | $7.54 \pm 0.27$ | $5.71 \pm 0.07$ |
| Mixup | $39.97 \pm 1.89$ | $29.62 \pm 1.54$ | $16.79 \pm 0.63$ | $10.47 \pm 0.48$ | $7.96 \pm 0.14$ |
| VAT | $8.41 \pm 1.01$ | $7.44 \pm 0.79$ | $5.98 \pm 0.21$ | $4.85 \pm 0.23$ | $4.20 \pm 0.15$ |
| MeanTeacher | $6.45 \pm 2.43$ | $3.82 \pm 0.17$ | $3.75 \pm 0.10$ | $3.51 \pm 0.09$ | $3.39 \pm 0.11$ |
| MixMatch | $3.78 \pm 0.26$ | $3.64 \pm 0.46$ | $3.27 \pm 0.31$ | $3.04 \pm 0.13$ | $2.89 \pm 0.06$ |

Table 6: Error rate (%) for SVHN.

 **B.3  SVHN+Extra**

 Training the same model with supervised learning on the entire 604388-example training set achieved
 an error rate of $1.71\%$.

| Methods/Labels | 250 | 500 | 1000 | 2000 | 4000 |
|---|---|---|---|---|---|
| PiModel | $13.71 \pm 0.32$ | $10.78 \pm 0.59$ | $8.81 \pm 0.33$ | $7.07 \pm 0.19$ | $5.70 \pm 0.13$ |
| PseudoLabel | $17.71 \pm 0.78$ | $12.58 \pm 0.59$ | $9.28 \pm 0.38$ | $7.20 \pm 0.18$ | $5.56 \pm 0.27$ |
| Mixup | $33.03 \pm 1.29$ | $24.52 \pm 0.59$ | $14.05 \pm 0.79$ | $9.06 \pm 0.55$ | $7.27 \pm 0.12$ |
| VAT | $7.44 \pm 1.38$ | $7.37 \pm 0.82$ | $6.15 \pm 0.53$ | $4.99 \pm 0.30$ | $4.27 \pm 0.30$ |
| MeanTeacher | $2.77 \pm 0.10$ | $2.75 \pm 0.07$ | $2.69 \pm 0.08$ | $2.60 \pm 0.04$ | $2.54 \pm 0.03$ |
| MixMatch | $2.22 \pm 0.08$ | $2.17 \pm 0.07$ | $2.18 \pm 0.06$ | $2.12 \pm 0.03$ | $2.07 \pm 0.05$ |

Table 7: Error rate (%) for SVHN+Extra.

Figure 4: Error rate comparison of MixMatch to baseline methods on SVHN+Extra for a varying
number of labels. With 250 examples we reach nearly the state of the art compared to supervised
training for this model.