[Reviews · NeurIPS 2019]

Reviewer 1



Originality: 7 Quality:8 Clarity: 4 Significance:7 Mixmatch combined a lot of classical extraordinary methods that used for semi-supervised learning and achieved state-of-the-art results by a large margin across many datasets and labeled data amounts. Compared to previous method, this method is not only a simple combination of different data augmentation methods and other methods, such as exponential model average (EMA), it also explores a path to fully combine the advantages of different methods. In short, this method is of course a big step for semi-supervised learning on image classification. However, the experiments on this paper still needs to be modified to be perfect and a fair comparison with previous paper, such as Mean-Teacher. Also, some small problems need to be fixed to be finally published. 1 Loss function design: In the code implementation of Mixmatch, it’s not a fixed lambda to combine two terms in loss function. However, it used a linear lambda change strategy to update lambda during training process. That’s not common in the loss function. I think further explanation of this design should be addressed. Also, it may be better to include a fixed lambda version as a comparison 2 The accuracy report is not reliable. In this paper, it shows the average accuracy of last 20 models, which did not make any sense. In this paper, we can’t get a final model that can be directly applied to the test set. That’s not reasonable. To be published, it’s not necessary for this paper to design a method to pick out a final model for testing without using validation set. 3 In this paper, it does not have a fair comparison with previous methods. Only take the previous state of art as an example. The paper does not compare with 13 layer conv-net in all the previous semi-supervised method. Compared to this structure, wide-resnet is more complex and may greatly influence the performance. Though this paper re-implemented the previous on the new backbone, it’s still not a fair comparison considering the fact that the author may not be familiar with the previous method as the original author. Therefore, I strongly suggest to compare with previous methods based on the 13-layer conv-net backbone on Cifar-10, Cifar100 and SVHN dataset. 4 To compare with the Mean Teacher on Cifar with Resnet backbone, I think it’s also better to include all the results with different number of labels instead of only with 4000 labels in Table 1. 5 For Cifar-100, the experiments are not enough to say that Mixmatch works pretty well on the dataset. First, you do not have the result with your original backbone wide-resnet. Second, cifar-100 is a more challenging task, thus it’s better to include your results with different number of labels to give the conclusion that MixMatch really makes a great difference to the semi-supervised classification area. Third, a fair comparison with Conv-Net is also necessary to compare with previous methods. 6 For the ablation study, I think it’s not enough and clear. Here, we want the clear ablation study to see the effect of different parts in Mixmatch. Personally, please include the results with the following suggestion: (1) MixMatch without EMA(only remove the EMA part) (2) Mixmatch without distribution averaging(only remove distribution average) (3) Mixmatch without temperature sharpening(only remove this in mixmatch) (4) Mixmatch without mixup(only remove this in mixmatch). From my experiments with my re-implementation, MixUp and EMA really contributed a lot to the final performance. Therefore, removing EMA is really necessary for a perfect ablation study. 7 ImageNet is the most popular dataset for semi-supervised learning to finally verify the performance. If you have time, experiments on Imagenet would help to verify the MixMatch’s stable performance for the semi-supervised image classification task. Post-rebuttal comments: The author’s response is nearly perfect in addressing all my concerns in my previous review. Here, I only have a small suggestion for you to be a final good paper. From the reproduction by my group, we found the EMA plays an essential role in achieving the results. Without it, there would be a non-unneglectable gap to the showed results. Therefore, it is encouraged to include an ablation study of the EMA to show its impact on the proposed model.

Reviewer 2



This work is a combination of mixup, Entropy Minimization(sharpening), Consistency Regularization, Traditional Regularization. These methods are developed long ago or recently, this paper well combined these methods. The method is well supported by experiment results and ablation study over each component. This paper is improtant to SSL field and should widely used in different area.

Reviewer 3



- It is hard to say this work has its own originality since it borrows core ideas from other methods like mixup or entropy-based methods. But this paper proposes a unified framework, which shows very good performance for semi-supervised learning, thus it provides a good baseline for further research. - The paper is well written and easy to understand. - The experiments are good. The proposed method was evaluated on various datasets with good ablation studies. - They provide source codes to understand and reproduce this work more easily. - The implementation details are a little bit unclear. For the example of line 159-164, it would be better to explain the detail of 'exponential moving average of parameters' and why the authors used this technique. - What happens when K (the number of data augmentations) is larger than 2? - The weight decay 0.02 seems to be much larger than the standard settings as 0.0025 in [1]. It would be better if the authors provide the reason for the large weight decay. - It would be better if the authors apply another regularizer (for example, cutmix [2], which is similar to implement as mixup, but shows better performance.) for the MixMatch framework. Overall, this paper is well written, the proposed framework makes sense, and the experimental results are good. So I want to give an accept for this paper. [1] Oliver, Avital, et al. "Realistic evaluation of deep semi-supervised learning algorithms." Advances in Neural Information Processing Systems. 2018. [2] Yun, Sangdoo, et al. "Cutmix: Regularization strategy to train strong classifiers with localizable features." arXiv preprint arXiv:1905.04899 (2019). ----------------------------------------------- The authors clarified some confusing things through the rebuttal. I want to keep my original rating (7).

[Author Response · NeurIPS 2019]

We thank the reviewers for their thorough comments, which we address below:

**Reviewer 1.** We ran additional experiments based on your detailed suggestions. Their results have been added to our manuscript and are summarized below.

**1. Loss function design.** We neglected to mention this fact in the paper: we linearly warm-up the lambda from 0 to its final value as previously done e.g. in [25, 34, 43]. Below we present the results with and without warmup:

| Method | CIFAR-10, 250 labels | CIFAR-10, 4000 labels |
|---|---|---|
| Without Warmup | 87.60 | 93.73 |
| With Warmup | **88.62** | **93.77** |

**2. Accuracy & Averaging.** The reason we report the average over the last checkpoints is to be more realistic and not use large validation sets (as discussed in [6, 34, 38]). If we simply return the model accuracy of the last trained checkpoint our results are consistent but have higher variance.

| Dataset | Labels | Method | Median of Last 20 | Accuracy of Last 5 Models | | | | |
|---|---|---|---|---|---|---|---|---|
| CIFAR-10 | 250 | Mean Teacher | 46.34 | 46.01 | 46.83 | 45.61 | 45.58 | 45.55 |
| CIFAR-10 | 250 | MixMatch | 85.60 | 85.55 | 85.65 | 85.62 | 85.75 | 85.68 |
| CIFAR-10 | 4000 | Mean Teacher | 88.47 | 88.43 | 88.55 | 88.60 | 88.61 | 88.54 |
| CIFAR-10 | 4000 | MixMatch | 93.16 | 93.07 | 93.12 | 93.05 | 93.14 | 93.14 |

**3. 13-Layer ConvNet.** As suggested we run experiments based on the TensorFlow Implementation of the 13-layer ConvNet from the Mean Teacher paper. MixMatch has a larger advantage when using this 13-layer network than when using the ResNet in our paper.

| Method | CIFAR-10 | | SVHN | |
|---|---|---|---|---|
| | 250 | 4000 | 250 | 1000 |
| Mean Teacher | 46.34 | 88.57 | 94.00 | 96.00 |
| **MixMatch** | **85.69** | **93.16** | **96.41** | **96.61** |

**4. Table 1 Comparison with Mean Teacher.** The purpose of Table 1 is to show the strongest reported results in prior papers along with our strongest MixMatch results. Unfortunately Mean Teacher does not report on CIFAR-100.

**5. CIFAR-100 Comparisons.** When new CIFAR-100 experiments finish we plan to include them in the final paper.

**6. Ablation Studies.** We believe we have included most of the ablations requested in Table 4 where we run MixMatch without MixUp (row 5), without sharpening (row 3), and without EMA (row 4). The best way to evaluate MixMatch without distribution averaging in our view is to set $K = 1$ augmentations (row 2). We address other values of $K$ below. We hope this clarifies the ablation studies we included; if you have other suggestions we would like to add them.

**7. ImageNet.** We also are excited for the possibility of MixMatch on ImageNet and hope to study it in future work.

**8. K=3, 4 Augmentations.** Thank you for the suggestion, also made by **R3**; we find that in practice $K = 2$ gives the best results for the least performance penalty. We included additional experiments in our revised manuscript which we also list below. Using $K > 1$ augmentations is necessary; further augmentations do not give as much of a gain.

| Dataset | $K = 1$ | $K = 2$ | $K = 3$ | $K = 4$ |
|---|---|---|---|---|
| CIFAR-10, 250 labels | 84.02 | **88.62** | 88.45 | 87.55 |
| CIFAR-10, 4000 labels | 92.00 | 93.73 | 93.77 | **94.12** |

**Reviewer 2.** To address your question, Equation (3) uses the standard cross-entropy loss. The reason we must move to a L2 loss in Equation (4) is to help stabilize the training process as previously identified in [25, 34].

**Reviewer 3.** As outlined in item 8 of our response to R1, we added a discussion of $K = 3$ or $K = 4$ augmentations in our revised manuscript. Thank you for bringing up the clarification for weight decay: we always multiply the weight decay value by the learning rate (2e-3 in all experiments) but neglected to mention this in the paper. Finally, we tried other regularization strategies (such as cutout) but found it gives inferior results. Cutmix may work even better and we hope that future work explores additional regularizations (e.g., cutmix, manifold mixup).

[Meta-Review · NeurIPS 2019]

The reviewers are in consensus that this is well-written paper, which combine a number of well-studied SSL methods. The results include good performance over a number of datasets. Thus the recommendation to accept this paper.